# Mean platelet volume and mean platelet volume to platelet count ratio as predictors of severity and mortality in sepsis

Jorge Luis Vélez-Páez[1,2], Pedro Legua[3☯], Pablo Vélez-Páez[4,5], Estefanía Irigoyen[2], Henry Andrade[5], Andrea Jara[6], Fernanda López[7], Jorge Pérez-Galarza[1,8☯], Lucy Baldeón🄳[1,8]*

1 Facultad de Ciencias Médicas, Universidad Central del Ecuador, Quito, Ecuador, 2 Centro de Investigación Clínica en Medicina Crítica, Hospital Pablo Arturo Suárez, Quito, Ecuador, 3 Instituto de Medicina Tropical Alexander von Humboldt, Universidad Peruana Cayetano Heredia, Lima, Perú, 4 Centro de Investigación Clínica en Medicina Crítica, Quito, Ecuador, 5 Unidad de Terapia Intensiva, Hospital General IESS de Ibarra, Imbabura, Ecuador, 6 Unidad de Emergencia, Hospital Básico de Machachi, Pichincha, Ecuador, 7 Instituto de Posgrado Medicina Crítica y Terapia Intensiva, Universidad Central del Ecuador, Quito, Ecuador, 8 Instituto de Investigación en Biomedicina, Universidad Central del Ecuador, Quito, Ecuador

☯ These authors contributed equally to this work.
* lybaldeon@uce.edu.ec

**Data Availability Statement:** All relevant data are within the manuscript and its Supporting Information files.

## Abstract

### Introduction

Sepsis is a public health problem due to its high prevalence and mortality. Mean platelet volume (MPV), a biomarker reported in routine blood counts, has been investigated and shows promise for determining fatal outcomes in septic patients.

### Objective

Evaluate whether the mean platelet volume (MPV) and mean platelet volume-to-platelet count (MPV/P) ratio are predictors of clinical severity and mortality in patients with sepsis.

### Methods

A prospective population cohort of 163 patients aged 18–97 years was recruited at the Intensive Care Unit of Pablo Arturo Hospital, Quito, Ecuador from 2017–2019 and followed up for 28 days. Patients were diagnosed with sepsis based on SEPSIS-3 septic shock criteria; in which the MPV and the MPV/P ratio were measured on days 1, 2, and 3. Sequential organ failure assessment (SOFA) score and presence of septic shock assessed clinical severity. Mortality on day 28 was considered the fatal outcome.

### Results

The average age of the patients was 61,15 years (SD 20,94) and female sex was predominant. MPV cutoff points at days 1, 2 and 3 were >9,45fL, >8,95fL and >8, 85fL; and (MPV/P) ratio >8, 18, >4, 12 y >3, 95, respectively. MPV at days 2 (9,85fL) and 3 (8,55fL) and (MPV/P) ratio at days 1 (4,42), 2 (4,21), and 3 (8,55), were predictors of clinical severity assessed

**Funding:** The funders had no role in study design, data collection, and analysis, decision to publish, or preparation of the manuscript. The Central University of Ecuador will cover the publication fees in case the manuscript be accepted.

**Competing interests:** The authors have declared that no competing interests exist.

by septic shock, which reached significance in the ROC curves. MPV and (MPV/P) ratio were also predictors of clinical severity determined by SOFA at days 1, 2, and 3, where higher values were observed in non-survivors reaching significance in all categories. MPV and MPV/P ratio at days 1, 2 and 3 were independent predictor factors of mortality using Cox proportional hazards model (HR 2,31; 95% CI 1,36–3,94), (HR 2,11; 95% CI 1,17–3,82), (HR 2,13; 95% CI 1,07–4,21) and (HR 2,38; 95% CI 1,38–4,12), (HR 2,15; 95% CI 1,14–4,06), (HR 4,43; 95% CI, 1,72–11,37) respectively.

## Conclusions

MPV and the MPV/P ratio are predictors of clinical severity and mortality in sepsis. The MPV and its coefficient are indicators of the biological behavior of platelets in sepsis. They should be considered as a cost-effective and rapidly available tool that guides the treatment.

## Introduction

Sepsis is a dysregulated response of the body to infectious disease [1], of high prevalence, and in which few measures have not been able to reduce its mortality, except for the early initiation of antibiotic therapy and adequate support for organ failure. It is considered a systemic process with a high probability of organic impact. The hemostatic system is not an exception and is frequently disturbed [2]. It has been shown that coagulation and platelet activation can occur in an early phase of this syndrome, playing a decisive role in its pathophysiology [3].

The reason why the platelet, and strictly the mean platelet volume (MPV) marked the biological behavior of sepsis is that the inflammation caused by infectious pathogens induces a systemic response with the production of pro-inflammatory cytokines, thrombopoietin, and other substances that stimulate the massive production of young platelets. These platelets are morphologically different in shape (spherical with pseudopods) and size (larger), but functionally poorly competent, inducing thrombogenic activity and adverse clinical outcomes [4, 5]. They activate the NLRP3 inflammasome in immune cells, which induces the expression of IL-1 [5, 6]. Platelets activated by immunoglobulin, thrombin, collagen, or phorbol myristate acetate (PMA) can also release free mitochondria that behave in a damage-associated molecular pattern that amplifies the inflammatory response [5, 7, 8]. In sepsis, a low platelet count is a well-known indicator of poor prognosis [9, 10]. The MPV measured in femtoliters (fL) and the coefficient (ratio) between the MPV and the absolute platelet count are anatomical biomarkers derived from platelets, rarely used, but reported in routine blood counts, are gaining interest as markers of evolution to multiple organ dysfunction, clinical severity, and mortality in septic states [9, 10]. The objective of this study is to demonstrate the usefulness of the MPV and mean platelet volume-to-platelet count (MPV/P) ratio as predictors of mortality and severity in patients with sepsis.

## Materials and methods

An observational analytical prospective study was carried out at the Intensive Care Unit of Pablo Arturo Suárez Hospital in Quito-Ecuador, to consecutively enroll patients with the diagnosis of sepsis from bacterial origin, from April 2017 to August 2019. Patients over 18 years of age with a diagnosis of sepsis under the SEPSIS-3 septic shock criteria were included [1]. Patients with non-infectious diseases that report increment of MPV such as acute ischemic

heart disease, acute embolic cerebrovascular disease, chronic hematological diseases; and who received platelets transfusion during hospitalization were excluded. From the clinical history, age, sex, site of infection, platelet count, leukocyte count, procalcitonin, and serum lactate were documented. The MPV and MPV/P ratio were measured upon admission to intensive care on days 1, 2, and 3. The clinical severity of sepsis was estimated by the presence of septic shock and multiple organ dysfunction (SOFA) score. The mortality outcome was documented on day 28. MPV was measured by impedance in a Siemens 21–20 machine, in venous blood collected in EDTA tubes.

Statistical analyzes were carried out with R software. Absolute and relative values of the qualitative variables, as well as measures of central tendency and variability for the quantitative variables, were reported. Bivariate analyzes were performed to compare the clinical character-istics and laboratory parameters between non-survivors and survivors. Chi-square test was applied for categorical variables, for quantitative variables Mann-Whitney test was used since they show significance in the normality tests (Kolmogorov-Smirnov). To compare MVP between day 1, day 2, and day 3, Friedman test of repeated measures (> 2 repetitions) was used. Survival analysis was performed using the Log Rank test (Mantel-Cox), and the Cox regression models as a multivariate analysis. Statistical significance was established if p was <0,05; the Hazard Ratio (HR) was considered significant observing 95% confidence interval, it was considered a risk factor if the lower limit was > 1. The research ethic committee of Uni-versidad Peruana Cayetano Heredia (Code N°: 100424) approved the development of the study. The ethics committee approved the waiver for clinical data collection of severely injured patients that we're unable to consent. Deferred written informed consent was approved for recovered patients; the documents were archived by the investigators. No minors were included in this study. Pablo Arturo hospital authorized the use of clinical patient information. The research was carried out in compliance with the internationally required ethical standards, respecting the confidentiality of the patients.

## Results

A total of 163 patients with sepsis diagnosis of bacterial origin were followed up for 28 days at the Intensive Care Unit (ICU). The average age was 61,15 years, with a predominance of females (61,96%). The mean ICU length of hospitalization stay of patients with sepsis was 6,40 days. This study compares a group of patients with sepsis of bacterial origin that did not sur-vive (34,36%) vs. a group of survivors (65,64%). The highest frequency of infection site was the gastrointestinal tract 44,17%, followed by the pulmonary 30,67%, urinary tract 15,95%, and other areas 9,20%. No differences related to infection sites were observed when comparing between groups. 74,69% of the patients received mechanical ventilation, observing significant differences between the non-surviving and surviving groups (94,55% vs. 64,49%; p = <0.00). 56,88% of the patients received sedation, observing significant differences between groups (81,48% vs 44,34%; p = <0,00) (Table 1).

When comparing the SOFA score between non-survivors and survivors, significant differ-ences were observed in the different evaluation stages on day 1 (10,68 vs. 7,99; p = <0,00), day 2 (10,30 vs. 7,19; p = <0,00) and day 3 (10,71 vs. 5,07; p = <0,00). The APACHE score did not show significant differences between groups. The proportion of patients with septic shock was 66,67%, finding significant differences when comparing non-survivors and survivors (82,14% vs. 58,88%; p = 0,00) respectively (Table 2).

The MVP at days 1, 2 and 3 were 9,11 fL, 8,95 fL and 9,05 fL respectively. Significant differ-ences were observed between non-survivors and survivors on day 1 (9,45 fL vs. 8,93 fl; p = 0,02), day 2 (9,42 fL vs. 8,33 fl; p = 0,01), and day 3 (9,50 fL vs. 8,88 fl; p = 0,02). In the

**Table 1. Clinical characteristics of patients attended with sepsis at 28 days of hospitalization comparing the hospital discharge condition.** Clinical Characteristics.

| | Total | Hospitalization discharge | | P-value |
|---|---|---|---|---|
| | | 28 days | | |
| | | No survivals | Survivals | |
| | | (n = 56) | (n = 107) | |
| Age, yr, mean ± SD[#] | 61,15 ± 20,94 | 65,68 ± 18,27 | 58,79 ± 21,92 | 0,06 |
| Gender, n (%)[$] | | | | |
| Male | 62 (38,04%) | 21 (37,50%) | 41 (38,32%) | 0,91 |
| Female | 101 (61,96%) | 35 (62,50%) | 66 (61,68%) | |
| SOFA, mean ± DE[#] | | | | |
| Day 1 | 8,92 ± 3,96 | 10,68 ± 3,61 | 7,99 ± 3,83 | <0,00** |
| Day 2 | 8,13 ± 3,75 | 10,30 ± 3,33 | 7,19 ± 3,54 | <0,00** |
| Day 3 | 6,65 ± 4,20 | 10,71 ± 3,78 | 5,07 ± 3,18 | <0,00** |
| APACHE at hospital admission, mean ± SD[#] | 18,92 ± 8,69 | 21,24 ± 9,61 | 17,72 ± 7,95 | 0,06 |
| Septic shock, n (%)[$] | | | | |
| Yes | 109 (66,87%) | 46 (82,14%) | 63 (58,88%) | 0,00* |
| No | 54 (33,13%) | 10 (17,86%) | 44 (42,12%) | |
| Site of infection, n (%)[$] | | | | |
| Gastrointestinal tract | 72 (44,17%) | 26 (46,43%) | 46 (42,99%) | 0,84 |
| Pulmonary | 50 (30,67%) | 18 (32,14%) | 32 (29,91%) | |
| Urinary tract | 26 (15,95%) | 7 (12,50%) | 19 (17,76%) | |
| Other | 15 (9,20%) | 5 (8,93%) | 10 (9,35%) | |
| Mechanical ventilation, n (%)[$] | | | | |
| Yes | 121 (74,69%) | 52 (94,55%) | 69 (64,49%) | <0,00** |
| No | 41 (25,31%) | 3 (5,45%) | 38 (35,51%) | |
| Sedation, n (%)[$] | | | | |
| Yes | 91 (56,88%) | 44 (81,48%) | 47 (44,34%) | <0,00** |
| No | 69 (43,13%) | 10 (18,52%) | 59 (55,56%) | |
| Length of hospital stay, mean ± SD[#] | 6,40 ± 4,71 | 6,07 ± 5,10 | 6,58 ± 4,51 | 0,06 |

\#: Mann-Whitney for independent samples

\$: Chi-square test

* significant differences p<0,05

** significant difference p<0,001.

SOFA: Sepsis-related Organ Failure Assessment. APACHE: Acute Physiology and Chronic Health Evaluation. SD: Standard deviation. n: number of cases.

platelet count, significant differences were observed between groups on day 1 (194,19 / mm3 vs. 231,46 / mm3 p = 0,04), day 2 (171,79 / mm3 vs. 221,95 / mm3 p = 0,02) and day 3 (159,32 / mm3 vs. 233,89 / mm3 p = 0,02). The mean of the MPV/P ratio at day 1, 2 and 3 was 7,62; 8,68 and 9,94 respectively. Significant differences were observed between non-survivors and survivors at day 1 (9,47 vs. 6,65; p = 0,02), day 2 (12,31 vs. 7,04; p = 0,00), and day 3 (15,74 vs. 8,88 fl; p = 7,77). Serum lactate presented a mean of 2,94 mmol / L, significant differences were observed between groups at day 1 (3,68 mmol / L vs 2,57 mmol / L; p = 0,00). Procalcitonin presented a mean of 30,51 ng / ml, the mean of SatO2VC was 65,85%, and the mean of leukocytes on day 1, 2 and 3 was 15,47 / mm3, 14,29 / mm3 and 13,92 / mm3 respectively. No significant differences were observed in these parameters when comparing between groups (Table 3).

**Table 2. Comparison of SOFA score base on ranges of values of MPV and MPV/platelets ratio.**

| Variables | SOFA | | | | | |
|---|---|---|---|---|---|---|
| | Day 1 | | Day 2 | | Day 3 | |
| | Mean ± SD | P-value | Mean ± SD | P-value | Mean ± SD | P-value |
| MPV Day 1 | | | | | | |
| ≥9,45 | 10,16 ± 3,97 | 0,00** | 9,59 ± 3,87 | <0,00** | 8,18 ± 4,68 | 0,00* |
| <9,45 | 8,04 ± 3,72 | | 7,15 ± 3,35 | | 5,73 ± 3,61 | |
| MPV Day 2 | | | | | | |
| ≥8,95 | 9,30 ± 3,86 | 0,03* | 8,91 ± 3,81 | 0,01* | 7,41 ± 4,34 | 0,02* |
| <8,95 | 8,04 ± 3,53 | | 7,40 ± 3,56 | | 5,97 ± 3,98 | |
| MPV Day 3 | | | | | | |
| ≥8,85 | 9,40 ± 3,70 | 0,00* | 8,99 ± 3,43 | 0,00* | 7,73 ± 4,34 | <0,00** |
| <8,85 | 7,80 ± 3,41 | | 7,11 ± 3,33 | | 5,24 ± 3,53 | |
| MPV/platelets Day 1 | | | | | | |
| ≥8,18 | 11,81 ± 3,74 | <0,00** | 10,48 ± 3,42 | <0,00** | 8,75 ± 4,13 | 0,00* |
| <8,85 | 8,10 ± 3,63 | | 7,58 ± 3,62 | | 6,20 ± 4,09 | |
| MPV/platelets Day 2 | | | | | | |
| ≥4,12 | 9,77 ± 3,70 | <0,00** | 9,10 ± 3,70 | <0,00** | 7,69 ± 4,36 | <0,00** |
| <4,12 | 7,11 ± 3,22 | | 6,80 ± 3,42 | | 5,16 ± 3,47 | |
| MPV/platelets Day 3 | | | | | | |
| ≥3,95 | 9,45 ± 3,71 | 0,00* | 8,81 ± 3,48 | 0,00* | 7,62 ± 4,51 | <0,00** |
| <3,95 | 7,18 ± 3,06 | | 6,78 ± 3,16 | | 4,71 ± 2,60 | |

* Significant differences p<0, 05

** significant difference p<0,001. SOFA: Sepsis-related Organ Failure Assessment. MPV: Mean Platelet Volume. SD: Standard deviation.

**Table 3. Comparison of laboratory parameters of patients with sepsis based on the condition of hospitalization discharge in survival or no survival.**

| Laboratory parameters | Total | Condition at hospital discharge | | P-value |
|---|---|---|---|---|
| | | No survival | Survival | |
| MPV, mean ± SD# fL | | | | |
| Day 1 | 9,11 ± 1,46 | 9,45 ± 1,46 | 8,93 ± 1,43 | 0,02* |
| Day 2 | 8,95 ± 1,39 | 9,42 ± 1,51 | 8,73 ± 1,28 | 0,01* |
| Day 3 | 9,05 ± 1,42 | 9,50 ± 1,54 | 8,88 ± 1,35 | 0,02* |
| Platelets/mm3, mean ± SD# | | | | |
| Day 1 | 218,66 ± 131,03 | 194,19 ± 130,96 | 231,46 ± 129,85 | 0,04* |
| Day 2 | 206,31 ± 131,41 | 171,79 ± 116,96 | 221,95 ± 135,08 | 0,02* |
| Day 3 | 213,61 ± 208,46 | 159,32 ± 101,79 | 233,89 ± 233,52 | 0,02* |
| MPV/platelets, mean ± SD# | | | | |
| Day 1 | 7,62 ± 10,74 | 9,47 ± 12,56 | 6,65 ± 9,58 | 0,02* |
| Day 2 | 8,68 ± 12,12 | 12,31 ± 16,36 | 7,04 ± 9,24 | 0,00* |
| Day 3 | 9,94 ± 17,83 | 15,74 ± 28,62 | 7,77 ± 10,93 | 0,01* |
| Leukocytes/mm3, mean ± SD# | | | | |
| Day 1 | 15,47 ± 11,32 | 15,53 ± 12,86 | 15,43 ± 10,49 | 0,37 |
| Day 2 | 14,29 ± 8,08 | 13,17 ± 7,89 | 14,81 ± 8,14 | 0,11 |
| Day 3 | 13,92 ± 9,01 | 13,69 ± 7,59 | 14,01 ± 9,52 | 0,99 |
| Lactate, mean ± SD# mmol/L | 2,94 ± 2,41 | 3,68 ± 2,97 | 2,57 ± 1,98 | 0,00* |
| ScvO2 at hospital admission, mean ± SD# % | 65,85 ± 13,53 | 67,70 ± 14,30 | 64,84 ± 13,09 | 0,36 |
| Procalcitonin, mean ± SD# ng/ml | 30,51 ± 46,20 | 34,94 ± 44,36 | 28,22 ± 47,19 | 0,12 |

#: Mann-Whitney for independent samples

$: Chi- square test

* significant differences p<0, 05. SD: Standard deviation. MVP; Mean platelet volume. ScvO2: Central Venous Oxygen Saturation.

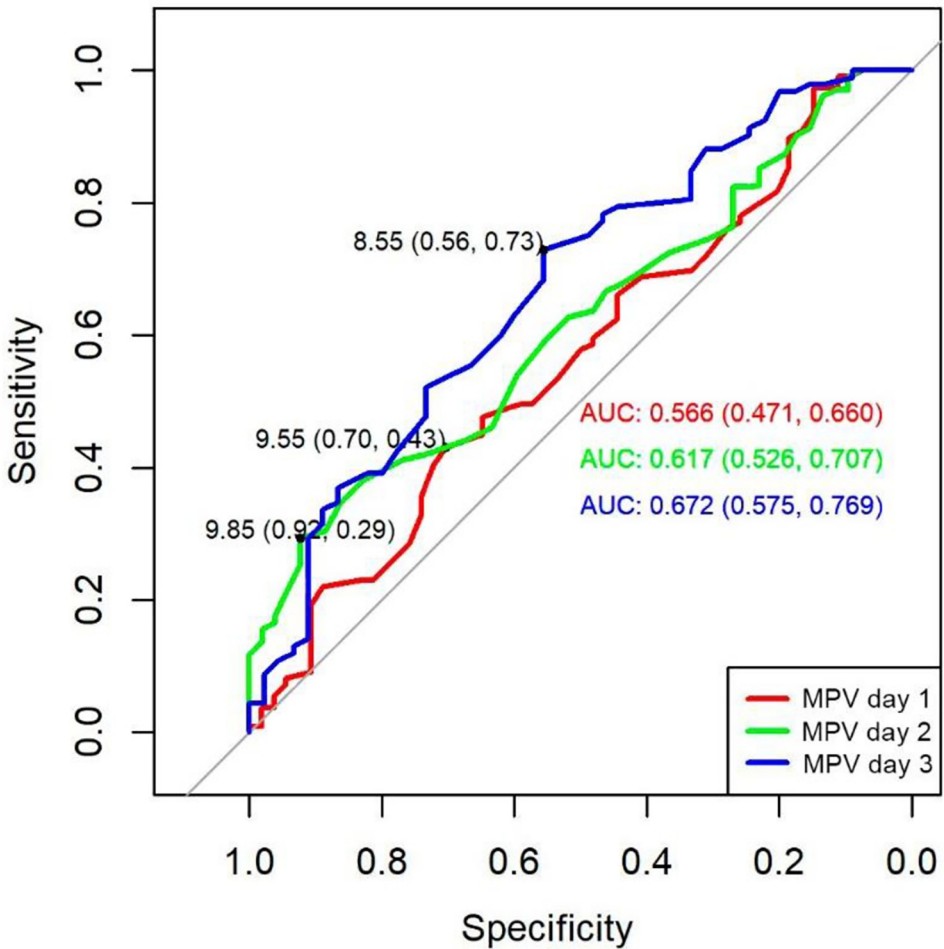

**Fig 1. ROC curve of MPV to predict septic shock in septic patients.** AUC: area under the curve.

## Severity and mortality predictors

ROC curve analysis of MPV did not predict septic shock at day 1 (AUC 0,56; 95% CI 0,47–0,66). However, at day 2 (AUC 0,61; 95% CI 0,52–0,70), and day 3 (AUC 0,67; 95% CI 0,57–0,76) ROC curve analysis were significant. The cut-off points for the MPV obtained by the Youden index were 9,85 for day 2 (specificity 92%, sensitivity 29%), for day 3 cut-off point was 8,55 (specificity 56%, sensitivity 73%). Although there were no differences between the curves, the one with the best performance corresponded to that of day 3 with a wider area (0,67) and a better relationship between specificity and sensitivity (Fig 1).

ROC curve analysis of (MPV/P) ratio significatively predicted septic shock at day 1 (AUC 0,59; 95% CI 0,50–0,68), day 2 (AUC 0,63; 95% CI 0,54–0,73), and day 3 (AUC 0,67; 95% CI 0,58–0,77). The cut-off points for MPV/P ratio obtained by Youden index were 4,42 for day 1 (specificity 65%, sensitivity 59%), for day 2 cut-off point was 4,21 (specificity 58%, sensitivity 65%), while for day 3 the cut-off point was 6,05 (specificity 80%, sensitivity 47%) (Fig 2).

ROC curve analysis of MPV predicted mortality at day 1 (AUC 0,60; 95% CI 0,51–0,69), day 2 (AUC 0,62; 95% CI 0,53–0,72), and day 3 (AUC 0,62; 95% CI 0,51–0,72) were significant. The cut-off points for the MPV obtained by Youden index were 9,45 for day 1 (specificity 66%,

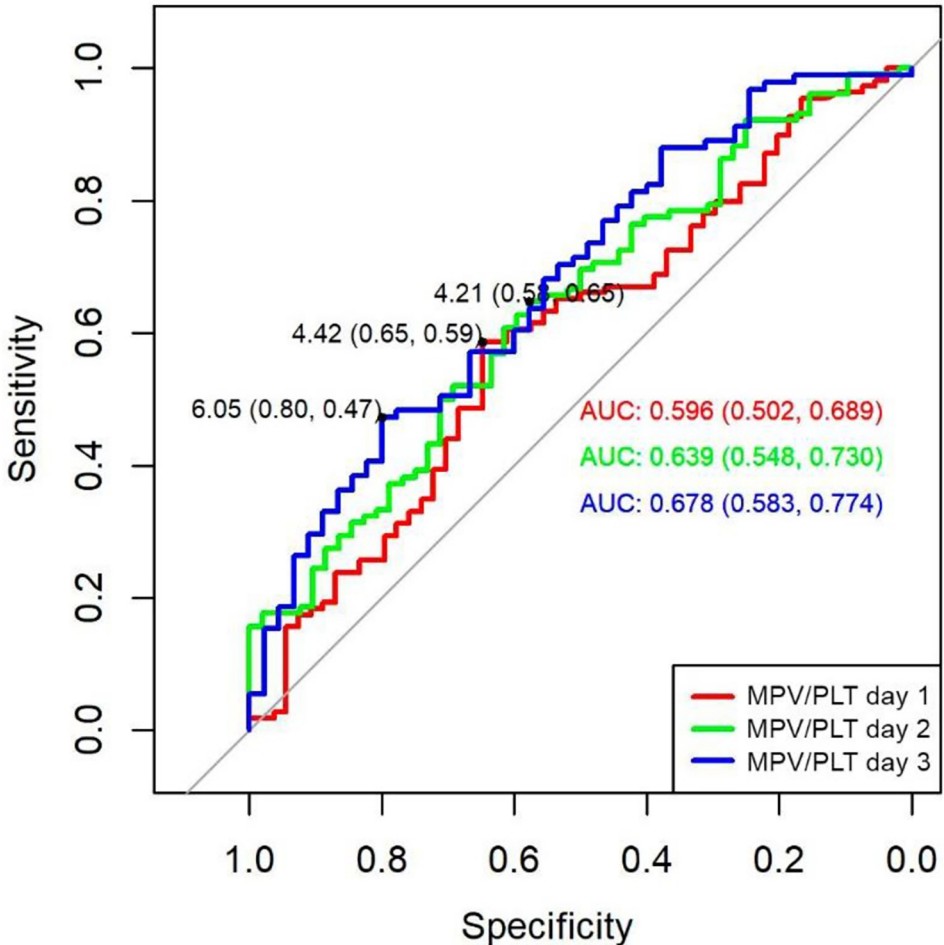

**Fig 2. ROC curve of MPV/P ratio to predict septic shock in septic patients.** AUC: area under the curve.

sensitivity 57%), for day 2 cut-off point was 8,95 (specificity 58%, sensitivity 65%), while for day 3 the cut-off point was 8,85 (specificity 54%, sensitivity 68%) (Fig 3).

ROC curve analysis of (MPV/P) ratio predicted mortality at day 1 (AUC 0,60 95% CI 0,51–0,70), day 2 (AUC 0,63; 95% CI 0,53–0,72) and day 3 (AUC 0,64; 95% CI 0,53–0,74). The cut-off points for MPV/P ratio obtained by Youden index were 8, 18 for day 1 (specificity 84%, sensitivity 36%), for day 2 cut-off point was 4,12 (specificity 48%, sensitivity 73%), while for day 3 the cut-off point was 3,95 (specificity 45%, sensitivity 86%). The confidence intervals of MPV and MPV/P ratio do not contain the value 0,5 therefore it is stated that the area under the ROC curve is significantly greater than the minimum required 0,5 for all days of evaluation (Fig 4).

In addition, the ROC curves for lactate and procalcitonin were performed. The results showed that serum lactate predicted mortality (AUC 0,62; 95% CI 0,53–0,71), the cut-off point was 2,85 (specificity 65%, sensitivity 56%). On the other hand, procalcitonin did not predict mortality (AUC 0,57; 95% CI 0,47–0,67) (S1 Fig).

## Survival analysis

In sepsis patients, the survival time evaluated was 28 days, where a clear tendency of decrease survival probability was observed. On day 1, survival was 95,10%, the highest number of deaths

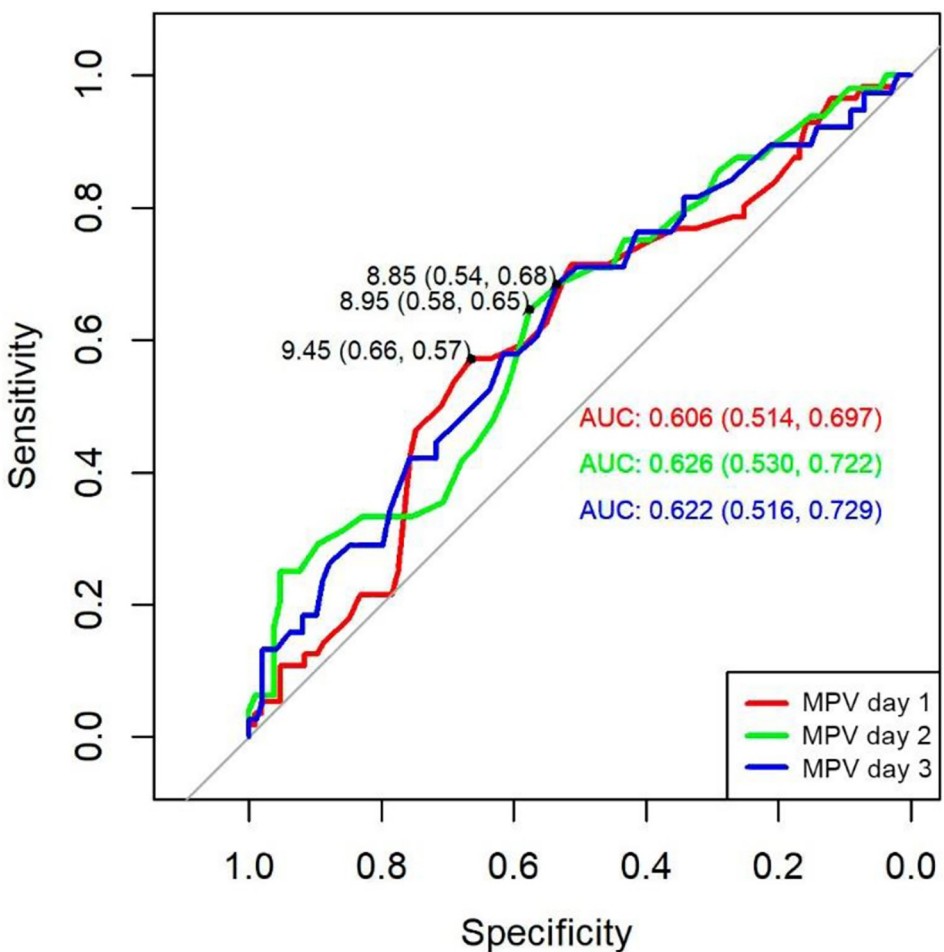

**Fig 3. ROC curve of MPV to predict the risk of mortality in septic patients.** AUC: area under the curve.

were observed in the first three days of follow-up (44,64%) while at the end of the follow-up (28 days) survival was 65,60%, there was no median survival for this time frame. Cut-off points of survival curves were compared for MPV on days 1, 2, and 3. When comparing the survival curves for the cut-off point of MPV ($\geq$9,45 and <9,45) on day 1, significant differences were observed (p = 0,00). A better behavior was observed for values <9,45. At a cut-off point of MPV ($\geq$8,95 and <8,95) on day 2, significant differences were observed (p = 0,01). A better behavior was observed for values <8,95. At a cut-off point of MPV ($\geq$8,85 and <8,85) on day 3, significant differences were observed (p = 0,02). A better behavior was observed for values <8,85 (Fig 5).

Cut-off points of survival curves were compared for MPV/P ratio on days 1, 2, and 3. When comparing the survival curves for the cut-off point of MPV/P ratio ($\geq$8,18 and <8,18) on day 1, significant differences were observed (p = 0,00). A better behavior was observed for values <8,18. Cut-off point of MPV ($\geq$4,12 and < 4,12) on day 2, significant differences were observed (p = 0,01). A better behavior was observed for values <4,12. Cut-off point of MPV ($\geq$3,95 and < 3,95) on day 3, significant differences were observed (p = <0,00). A better behavior was observed for values <3,95 (Fig 6).

For all variables, the Cox regression model was carried out in a crude and unadjusted way because of multicollinearity. Lactate was excluded from this analysis, which did not comply

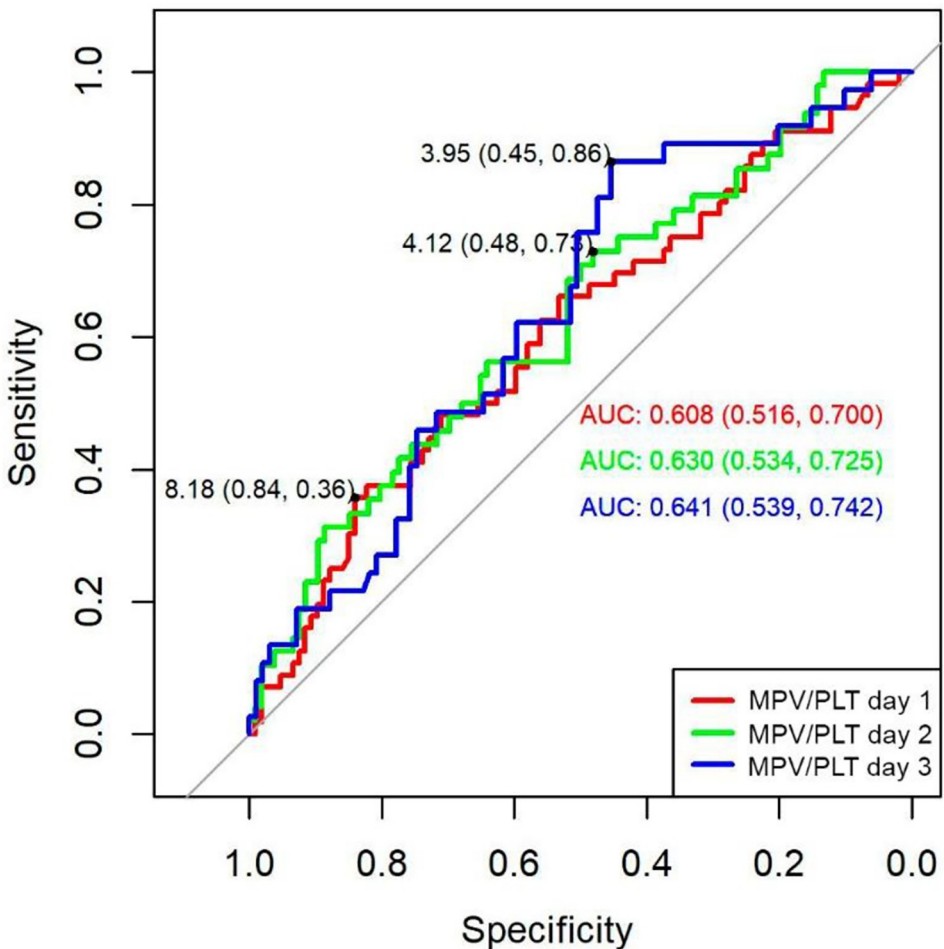

**Fig 4. ROC curve of MPV/P ratio to predict the risk of mortality in septic patients.** AUC: area under the curve.

with the proportional hazard assumption. All variables were significant (p = <0,05) for survival prognosis; MPV at day 1 ≥ 9,45, day 2 ≥8,95, and day 3 ≥8,85 presented a risk of 2,31; 2,11 and 2,13 times higher for those who presented values lower than the cut-off point. On the other hand, the MPV/P ratio at day 1 ≥8,18; day 2 ≥4,12 and day 3 ≥3,95 presented a risk of 2,38; 2,15 and 4,43 times higher for those who presented values lower than the cut-off point (Table 4).

## Discussion

The results indicate that the MPV on days 1 (cut-off ≥9,45), day 2 (cut-of ≥8,95), and day 3 (cut-off ≥8,85), were associated with mortality presenting a risk of 2,31; 2,11 and 2,13 times higher for not surviving respectively. MPV/P ratio on days 1 (cut-off ≥8,18), day 2 (cut-off ≥4,12), and day 3 (cut-off ≥3,95) were associated with mortality presenting a risk of 2,38; 2,15 and 4,43 times higher for not surviving respectively.

Several studies have demonstrated the predictive utility of MPV in sepsis [4, 11–16], Vardon-Bounes F, et al [17], reported MPV as a predictor of mortality (HR = 3,79) at 90-day survival study. Additionally, a meta-analysis performed by Tajarernmuang P. et al. [18], reported 11 studies (n = 3724), in which a significant association was determined between the MPV and

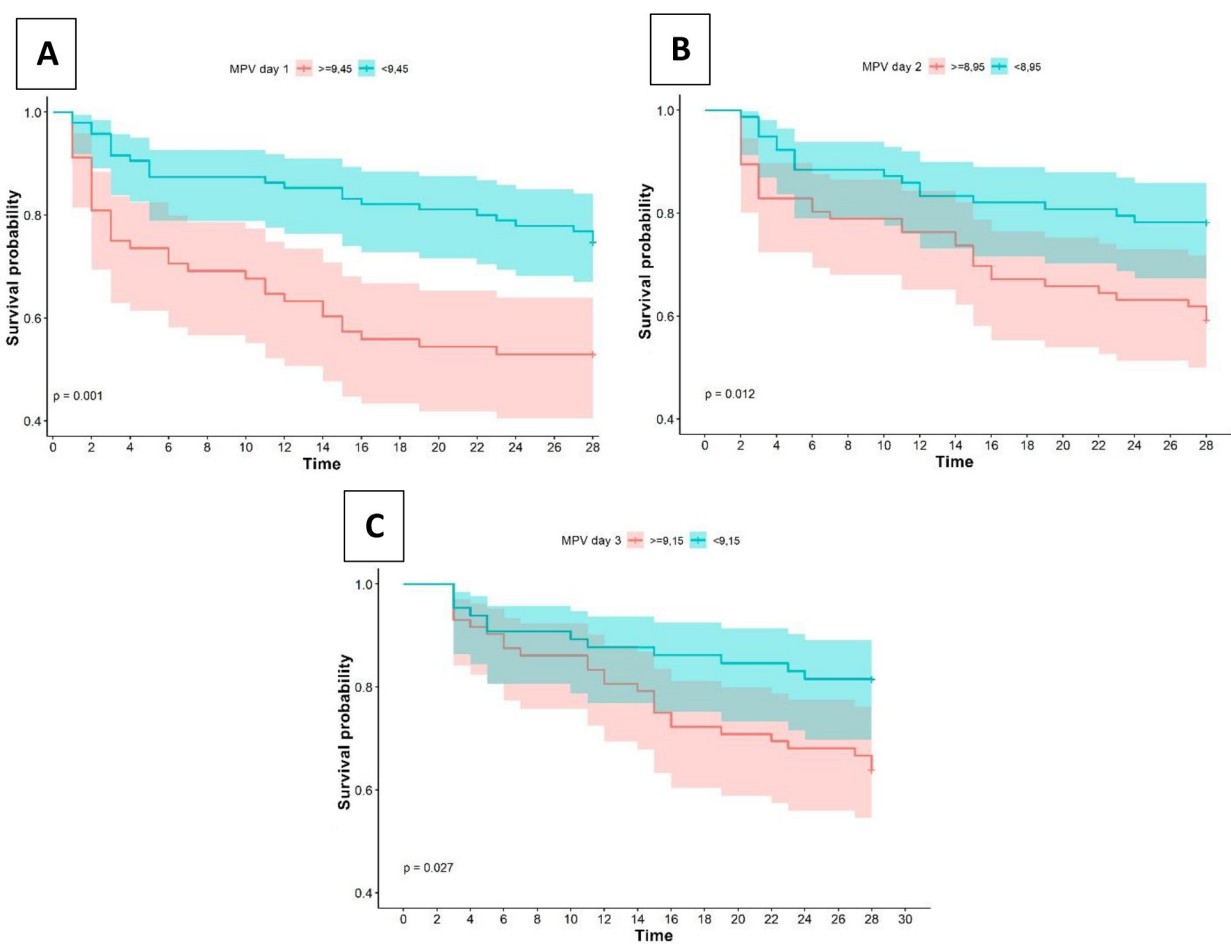

**Fig 5.** Cumulative survival curve of septic patients by MPV cut-off at day 1 (A); 2 (B); 3 (C).

mortality in critically ill patients on day 3. In our study, the MPV in all-time points was higher in no survivors. Cut-off points of $> 9,45$fL,$> 8,95$fL and$> 8,85$fL were predictors of mortality. Because the kinetics of MPV in sepsis is analogous to the response to antibiotic treatment; which tends to control the systemic infection between 48 to 72 hours of management, in this study we carried out the follow-up during the first 72 hours.

Despite the failure of Ates et al to determine that MPV/P ratio was a good estimator of mortality in sepsis [2]; in our study, the ratios of day 1 ($> 8,18$), 2 ($> 4,12$), and day 3 ($> 3,95$); were significant in the survival curves and were predictive factors of mortality using Cox regression model. This shows the predictive capacity of this marker and confirms its usefulness. Oh et al in 2016, found the predictive utility of MPV/P ratio at admission and 24 hours with both significant HR [19]. This is explainable since the ratio depends on the MPV (numerator) and the platelet count (denominator); then, the larger the platelet and the more marked thrombocytopenia, the more severe sepsis and the higher ratio.

Estimating clinical severity given by the presence of septic shock, the MPV of day 2 (9,85fL) and day 3 (8,55fL) and the MPV/P ratio of day 1 (4,42), day 2 (4,21), and day 3 (8,55), were moderate predictors that reached significance in the ROC curves. Regarding the clinical severity established by multi-organ dysfunction measured with SOFA, where, the MPV and the MPV/P ratio from days 1 to 3 were always higher in non-survivors with significance in all

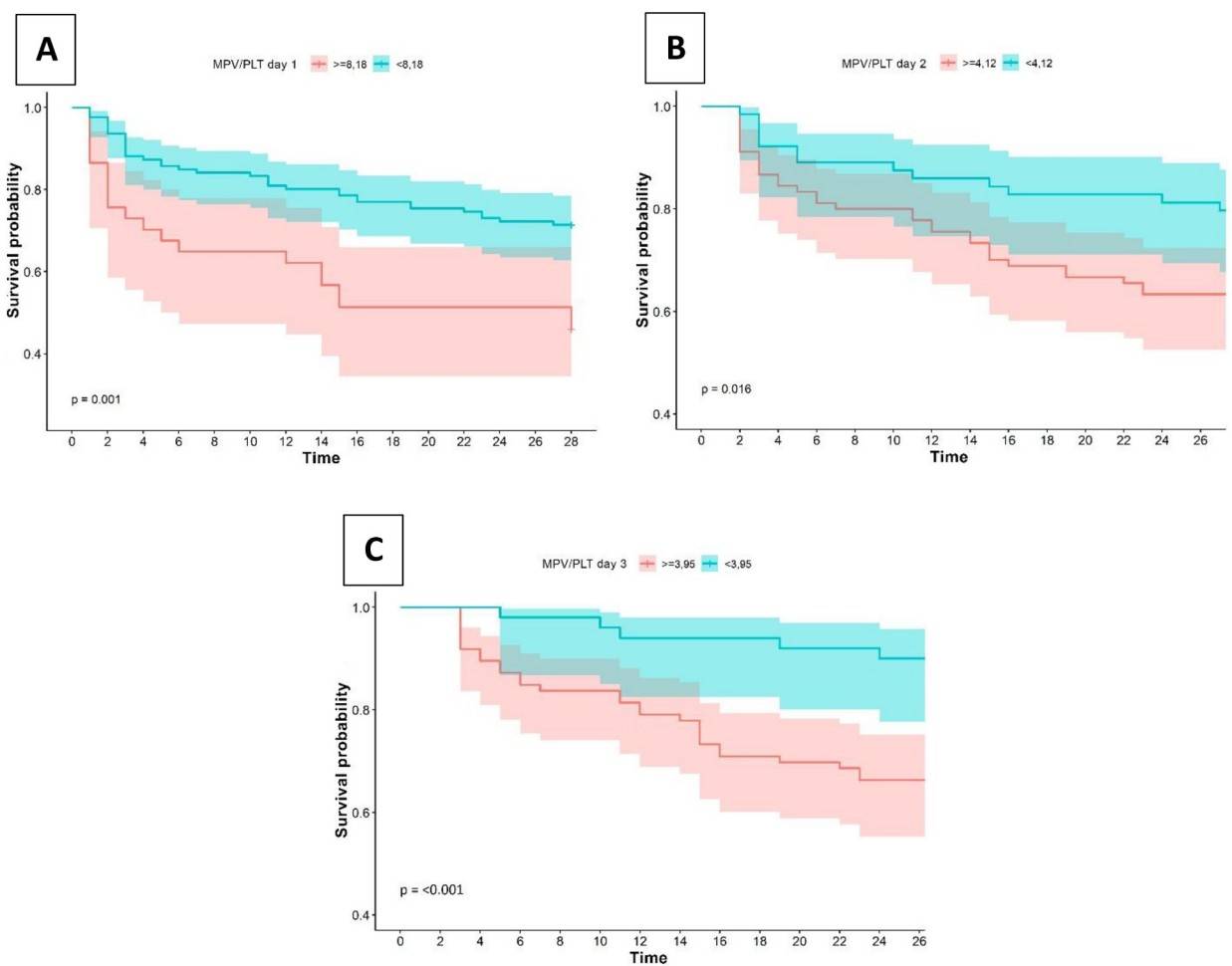

**Fig 6.** Cumulative survival curve of septic patients by MPV/P ratio cut-off at day 1 (A); 2 (B); 3 (C).

categories, this indicated that platelet biomarkers are good biological estimators of severity in sepsis. Similar findings were reported in other studies [20, 21].

The relevance of our findings lies in the fact that the MPV is reported in routine blood counts and its ratio is calculated with data from the same lab test, with low cost and global availability, making it a profitable anatomical biomarker with immediate clinical applicability, especially in limited-resource sites.

**Table 4. Cox regression for the survival of patients with sepsis.**

| Variables | β | Hazard Ratio (HR) | HR 95% CI | p-value | Proportional test risk p-value |
|---|---|---|---|---|---|
| MPV Day 1 ≥9,45 | 0,84 | 2,31 | 1,36–3,94 | 0,00* | 0,09* |
| MPV Day 2 ≥8,95 | 0,75 | 2,11 | 1,17–3,82 | 0,01* | 0,89* |
| MPV Day 3 ≥8,85 | 0,75 | 2,13 | 1,07–4,21 | 0,03* | 0,30* |
| MPV/P Day 1 ≥8,18 | 0,87 | 2,38 | 1,38–4,12 | 0,00* | 0,41* |
| MPV/P Day 2 ≥4,12 | 0,77 | 2,15 | 1,14–4,06 | 0,01* | 0,86* |
| MPV/P Day 3 ≥3,95 | 1,49 | 4,43 | 1,72–11,37 | 0,00* | 0,62* |

*significant differences p<0, 05. SD: Standard deviation. MVP: Mean platelet volume.

## Conclusions

The mean platelet volume (MPV) and mean platelet volume-to-platelet count ratio (MPV/P) are predictors of clinical severity and mortality in sepsis in their serial measurements in the first 72 hours. If the value of MPV and MPV/P ratio has not decreased within 72 hours after adequate antibiotic treatment, the mortality is high. MPV and MPV/P ratio are biomarkers of low cost, therefore they should be considered as tools that guide the treatment of sepsis, fundamentally in low resources settings.

## Limitations

A limitation of the study is that MPV was measured with impedance and used EDTA as an anticoagulant. EDTA increases the size of platelets in the post-analytical phase [10], however, when using K3 tubes, which contain little EDTA, this effect is minimized, added to the fact that the intensive therapy samples were processed within the first 15 minutes of receipt, which decreased the measurement bias. It should be remembered that Dastjerdi et al, found an adequate correlation of the MPV measured with EDTA compared to citrate, which further improves the reliability of the results [22]. Another controversial point is that in similar research studies and our work, high MPV was associated with severity and mortality, but there is not a generalized cut-off point, this is partly explained by the heterogeneity in techniques and anticoagulants used. Perhaps there is a racial component, since the values reported in eastern countries, specifically in China, show cut-off points of MPV higher than 11 [16, 23] compared to western countries. Furthermore, the study was carried out in a center located at 2,800 meters above sea level. Studies report that at high altitudes, hypoxia in non-acclimatized people generates hyperreactivity and greater platelet aggregation in response to adenosine diphosphate (ADP) and could increase MPV [24]. Evidence report MPV greater than 8.7 to predict adverse outcomes [18, 25, 26]. In our study, the MPV values exceeded this estimate throughout the follow-up. Besides, this study considers only the first 3 days of monitoring MPV. Extending the period of observation to the whole period of the hospitalization of the patients may be very useful and prompts us to carry out longer follow-up in subsequent studies.

## Supporting information

**S1 Fig. Roc curve of serum lactate and procalcitonin.** Serum lactate and procalcitonin Roc curve to predict the risk of mortality in septic patients.
(TIF)

## Author Contributions

**Conceptualization:** Jorge Luis Vélez-Páez, Pedro Legua, Pablo Vélez-Páez.

**Data curation:** Jorge Luis Vélez-Páez, Pedro Legua, Pablo Vélez-Páez, Estefanía Irigoyen, Henry Andrade, Andrea Jara, Fernanda López.

**Formal analysis:** Jorge Luis Vélez-Páez, Pedro Legua, Estefanía Irigoyen, Henry Andrade, Andrea Jara.

**Funding acquisition:** Jorge Luis Vélez-Páez, Lucy Baldeón.

**Investigation:** Fernanda López.

**Methodology:** Jorge Luis Vélez-Páez, Pedro Legua, Pablo Vélez-Páez, Estefanía Irigoyen, Andrea Jara, Fernanda López, Jorge Pérez-Galarza, Lucy Baldeón.

**Resources:** Jorge Pérez-Galarza.

**Software:** Jorge Luis Vélez-Páez, Pablo Vélez-Páez, Estefanía Irigoyen, Henry Andrade.

**Supervision:** Jorge Luis Vélez-Páez, Lucy Baldeón.

**Validation:** Jorge Luis Vélez-Páez, Henry Andrade, Jorge Pérez-Galarza, Lucy Baldeón.

**Visualization:** Jorge Luis Vélez-Páez, Jorge Pérez-Galarza, Lucy Baldeón.

**Writing – original draft:** Jorge Luis Vélez-Páez, Pedro Legua, Pablo Vélez-Páez, Estefanía Irigoyen, Henry Andrade, Andrea Jara, Fernanda López, Jorge Pérez-Galarza, Lucy Baldeón.

**Writing – review & editing:** Jorge Luis Vélez-Páez, Jorge Pérez-Galarza, Lucy Baldeón.

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
