## [Decision Letter · Decision Letter 0]

22 Jun 2021

PONE-D-21-06833

Mean Platelet Volume and Mean Platelet Volume to Platelet Count Ratio as Predictors of Severity and Mortality in Sepsis

PLOS ONE

Dear Dr. Baldeón,

Thank you for submitting your manuscript to PLOS ONE. After careful consideration, we feel that it has merit but does not fully meet PLOS ONE’s publication criteria as it currently stands. Therefore, we invite you to submit a revised version of the manuscript that addresses the points raised during the review process.

We look forward to receiving your revised manuscript.

Kind regards,

Andrea Ballotta

Academic Editor

PLOS ONE

Journal Requirements:

2. Thank you for stating in the text of your manuscript "Waiver of consent for clinical data collection of severe injured patients that were unable to consent was requested. Deferred consent from surviving patients was requested."

Please clarify whether the waiver of consent and deferred consent were approved.

Please also ensure that you have specified:

 - whether consent was informed

 - what type of consent you obtained (for instance, written or verbal, and if verbal, how it was documented and witnessed).

 - if your study included minors, state whether you obtained consent from parents or guardians.

'The funders had no role in study design, data collection and analysis, decision to publish, or preparation of the manuscript.'

5. Your abstract cannot contain citations. Please only include citations in the body text of the manuscript, and ensure that they remain in ascending numerical order on first mention.

6. Please include a separate caption for each figure in your manuscript.

7. Your ethics statement should only appear in the Methods section of your manuscript. If your ethics statement is written in any section besides the Methods, please move it to the Methods section and delete it from any other section. Please ensure that your ethics statement is included in your manuscript, as the ethics statement entered into the online submission form will not be published alongside your manuscript.

Additional Editor Comments:

Dear authors

on the basis of the reviewer's comments i deem the paper Mean Platelet Volume and Mean Platelet Volume to Platelet Count Ratio as Predictors of Severity and Mortality in Sepsis not suitable for publication. A major revision has to be performed.

Reviewers' comments:

Reviewer's Responses to Questions

**Comments to the Author**

1. Is the manuscript technically sound, and do the data support the conclusions?

Reviewer #1: Partly

2. Has the statistical analysis been performed appropriately and rigorously? 

Reviewer #1: I Don't Know

3. Have the authors made all data underlying the findings in their manuscript fully available?

Reviewer #1: Yes

4. Is the manuscript presented in an intelligible fashion and written in standard English?

Reviewer #1: Yes

5. Review Comments to the Author

Reviewer #1: The aim of the authors is to assess the value of MPV and MPV to platelet count ratio as predictors of severity and mortality in patients affected by sepsis. This has been previously done in various series of patients and clinical settings, and in sepsis in particular. Nevertheless, the manuscript may be of clinical interest if properly revised in order to highlight potential points of interest.

In particular, I would to raise some major issues concerning the content of the manuscript:

- the authors consider sepsis as all the same event, without giving detail whether it is of bacteric vs. viral origin, nor describe in detail the infective agents. A stratification of patients based on the nature of the infection and assessment of potential difference in influencing MPV and MPV to platelet count ratio may increase the value of the works.

- the authors consider only the first 3 days for MPV monitoring (as the most published reports do). Extending the period of observation to the whole period of the hospitalization of the patients, assessing the dynamics and tracing the trajectory of MPV may also be very useful in understanding this condition and constitute the very point of novelty of the works.

- grammar and syntaxis must be thoroughly revised: MPV is often reported as VMP, numbers are reported with 1 to 3 decimals, number are given with "." or "," .

6. PLOS authors have the option to publish the peer review history of their article (what does this mean?). If published, this will include your full peer review and any attached files.

Reviewer #1: No

---

## [Author Response · Author response to Decision Letter 0]

4 Aug 2021

Review Comments to the Author

Reviewer #1: The aim of the authors is to assess the value of MPV and MPV to platelet count ratio as predictors of severity and mortality in patients affected by sepsis. This has been previously done in various series of patients and clinical settings, and in sepsis in particular. Nevertheless, the manuscript may be of clinical interest if properly revised in order to highlight potential points of interest. 

Response: 

Dear reviewer thank you for your comment and for allowing us to review the manuscript. We agree with you, some studies evaluate the behavior of the mean platelet volume in several clinical scenarios and sepsis in particular. However, this study concerns interest, since it was carried out in Quito, a high-altitude city (2850 MASL), where the hypobaric hypoxia increases platelet reactivity. This could induce a prothrombotic phenotype determining an unusual behavior of the platelet indices. (Rocke AS, Paterson GG, Barber MT, Jackson AIR, Main S, Stannett C, et al. Thromboelastometry and platelet function during acclimatization to high altitudes. Thrombi Haemost. 2018; 118 (1): 63–71).

In particular, I would to raise some major issues concerning the content of the manuscript:

- the authors consider sepsis as all the same event, without giving detail whether it is of bacteric vs. viral origin, nor describe in detail the infective agents. A stratification of patients based on the nature of the infection and assessment of potential difference in influencing MPV and MPV to platelet count ratio may increase the value of the works.

Response: 

The study was carried out only in patients with sepsis of bacterial etiology, with diverse infectious foci such as pulmonary, abdominal, urinary, and others; the site of infection was not associated with mortality in the bivariate statistical analysis, so we did not perform subgroup analysis. This was stated and clarified in the manuscript (lines 82, 120-127). Unfortunately, we did not have access to the data to describe in detail the infective agents; this is a limitation of our study. 

- the authors consider only the first 3 days for MPV monitoring (as the most published reports do). Extending the period of observation to the whole period of the hospitalization of the patients, assessing the dynamics, and tracing the trajectory of MPV may also be very useful in understanding this condition and constitute the very point of novelty of the works.

Response: 

The kinetics of MPV in sepsis parallels the response to the antibiotic treatment administered; that is, it tends to decrease in patients in whom the systemic infection has been controlled and this is observed between 48 to 72 hours of management. For this reason, follow-up during the initial 72 hours is important (lines 249-252). Most studies on this subject, like ours, measure the MPV in the first hours after admission. However, we agree with you, few studies followed up to 15 days after the patients were discharged from intensive care and evaluated 90-day mortality. For example, Vardon Bounes F, Gratacap MP, Groyer S, et al. Kinetics of mean platelet volume predicts mortality in patients with septic shock. PLoS One. 2019; 14 (10): e0223553. Published 2019 Oct 17. doi: 10.1371 / journal.pone.0223553; that found that patients with MPV > 11.6 fL at day 10 of follow-up did not survive (HR: 3.79). This highlights the usefulness of this biomarker both in the critical unit and in hospitalization to determine the prediction of important outcomes such as mortality and prompts us to carry out longer follow-up in subsequent studies. We include this important observation in the manuscript (lines 299-302).

- grammar and syntaxis must be thoroughly revised: MPV is often reported as VMP, numbers are reported with 1 to 3 decimals, the number is given with "." or ",".

Response: 

We checked the grammar and syntaxis, the abbreviations are maintained as MPV, numbers reported with 2 decimals, and number are given with "," in all the manuscript.

---

## [Decision Letter · Decision Letter 1]

22 Dec 2021

Mean Platelet Volume and Mean Platelet Volume to Platelet Count Ratio as Predictors of severity and mortality in Sepsis

PONE-D-21-06833R1

Dear Dr. Baldeón,

We’re pleased to inform you that your manuscript has been judged scientifically suitable for publication and will be formally accepted for publication once it meets all outstanding technical requirements.

Kind regards,

Andrea Ballotta

Academic Editor

PLOS ONE

Additional Editor Comments (optional):

The manuscript is ready for publication

Reviewers' comments:

Reviewer's Responses to Questions

**Comments to the Author**

1. If the authors have adequately addressed your comments raised in a previous round of review and you feel that this manuscript is now acceptable for publication, you may indicate that here to bypass the “Comments to the Author” section, enter your conflict of interest statement in the “Confidential to Editor” section, and submit your "Accept" recommendation.

Reviewer #1: All comments have been addressed

2. Is the manuscript technically sound, and do the data support the conclusions?

Reviewer #1: Yes

3. Has the statistical analysis been performed appropriately and rigorously? 

Reviewer #1: I Don't Know

4. Have the authors made all data underlying the findings in their manuscript fully available?

Reviewer #1: Yes

5. Is the manuscript presented in an intelligible fashion and written in standard English?

Reviewer #1: Yes

6. Review Comments to the Author

Reviewer #1: No further comments from my side. My comments have been addressed thoroughly and extensively. I thank the authors for the consideration they gave to my observations.

7. PLOS authors have the option to publish the peer review history of their article (what does this mean?). If published, this will include your full peer review and any attached files.

Reviewer #1: No

---

## [Editor Report · Acceptance letter]

28 Dec 2021

PONE-D-21-06833R1 

Mean Platelet Volume and Mean Platelet Volume to Platelet Count Ratio as predictors of severity and mortality in Sepsis 

Dear Dr. Baldeón:

I'm pleased to inform you that your manuscript has been deemed suitable for publication in PLOS ONE. Congratulations! Your manuscript is now with our production department. 

Kind regards, 

on behalf of

Dr. Andrea Ballotta 

Academic Editor

PLOS ONE